# Health-Related Quality of Life and Satisfaction in Atrial Fibrillation Patients on Anticoagulant Therapy: Differences between Vitamin K Antagonists and Direct Oral Anticoagulants; Results from the Multicentre REGUEIFA Registry

**DOI:** 10.3390/jcm13175283

**Published:** 2024-09-06

**Authors:** Pilar Cabanas-Grandío, Laila González-Melchor, María Vázquez Caamaño, Emiliano Fernández-Obanza Windcheid, Eva González Babarro, Olga Durán Bobín, Miriam Piñeiro Portela, Oscar Prada Delgado, Juliana Elices Teja, Mario Gutiérrez Feijoo, Evaristo Freire, Oscar Díaz Castro, Javier Muñiz García, Javier García-Seara, Carlos González-Juanatey

**Affiliations:** 1Cardiology Department, Hospital Álvaro Cunqueiro, 36312 Vigo, Spain; maria.pilar.cabanas.grandio@sergas.es (P.C.-G.); odiazcastro@hotmail.com (O.D.C.); 2Cardiology Department, Hospital Clínico de Santiago de Compostela, 15706 A Coruña, Spain; laila.gonzalez.melchor@sergas.es (L.G.-M.); javiergarciaseara@yahoo.es (J.G.-S.); 3Cardiology Department, Hospital Povisa, 36211 Vigo, Spain; macrisvc@yahoo.es; 4Cardiology Department, Hospital Arquitecto Marcide, 15405 Ferrol, A Coruña, Spain; emiliano.patricio.fernandez-obanza.windscheid@sergas.es; 5Cardiology Department, Hospital de Montecelo, 36071 Pontevedra, Spain; eva.gonzalez.babarro@sergas.es; 6Cardiology Department, Hospital Universitario Lucus Augusti, 27002 Lugo, Spain; odbobin@gmail.com (O.D.B.); juliana.elices.teja@sergas.es (J.E.T.); 7Cardiology Department, Complexo Hospitalario Universitario de A Coruña, Instituto de Investigación Biomédica de A Coruña (INIBIC), 15006 A Coruña, Spain; miriam.pineiro.portela@sergas.es (M.P.P.); oscar.prada.delgado@sergas.es (O.P.D.); 8Cardiology Department, Hospital de Ourense, 32005 Ourense, Spain; mario.gutierrez.feijoo@sergas.es (M.G.F.); evaristo.freire.castroseiros@sergas.es (E.F.); 9Odds S.L., 15005 A Coruña, Spain; javmu@udc.es

**Keywords:** quality of life, patient satisfaction, atrial fibrillation, oral anticoagulation

## Abstract

**Background**: Oral anticoagulation (OAC) is pivotal in the clinical management of atrial fibrillation (AF) patients. Vitamin K antagonists (VKAs) and direct oral anticoagulants (DOACs) prevent thromboembolic events, but information about the quality of life (QoL) and patient satisfaction in relation with the anticoagulant treatment is limited. **Methods**: REGUEIFA is a prospective, observational, and multicentre study that included patients with AF treated by cardiologists. We included patients treated with VKAs or DOACs. The EuroQol-5D (EQ-5D) questionnaire evaluated QoL, and the Anti-Clot Treatment Scale (ACTS) questionnaire investigated patient satisfaction with OAC. **Results**: A total of 904 patients were included (532 on VKA and 372 on DOACs). A total of 846 patients completed the EQ-5D questionnaire, with results significantly worse in patients on VKAs than on DOACs: more mobility limitations (37.6% vs. 24.2%, *p* < 0.001), more restriction in usual activities (24.7% vs. 18.3%, *p* = 0.026), more pain/discomfort (31.8% vs. 24.2%, *p* = 0.015), a lower visual analogue scale (VAS) score (66.4 ± 16.21 vs. 70.8 ± 15.6), and a lower EQ-D5 index (0.79 ± 0.21 vs. 0.85 ± 0.2, *p* < 0.001). After adjusting for baseline characteristics, VKA treatment was not an independent factor towards worse EQ-5D results. Also, 738 patients completed the ACTS questionnaire, and burden and profit scores were lower in patients on VKAs than for DOACs (52.1 ± 8.4 vs. 55.5 ± 6.8, *p* < 0.001 and 11.1 ± 2.4 vs. 11.8 ± 2.6, *p* < 0.001, respectively). The negative impact score was higher for VKAs than for DOACs (1.8 ± 1.02 vs. 1.6 ± 0.99, *p* < 0.001), with a general positive impact score lower for VKAs than for DOACs (3.6 ± 0.96 vs. 3.8 ± 1.02, *p* < 0.001). **Conclusions**: Patients on VKA have more comorbidity and worse EQ-5D and VAS scores than those on DOACs. VKA has a greater burden and higher negative impact on the patient’s life than DOACs.

## 1. Introduction

Atrial fibrillation (AF) is the most common arrhythmia in clinical practice and is associated with increased mortality and morbidity, mainly due to stroke and heart failure [1,2,3]. Anticoagulant treatment with vitamin K antagonists (VKAs) or direct oral anticoagulants (DOACs) are safe and effective for the prevention of stroke and systemic embolism in patients with AF [1,2,3,4,5]. Also, DOACs show a favourable balance between safety and efficacy compared with VKA [5,6]. However, anticoagulant treatment has been associated with a poorer quality of life (QoL) in these patients [7,8,9,10]. The impact of oral anticoagulation in terms of QoL and patient satisfaction is not clearly characterised, as data about the role of different oral anticoagulants in QoL are scarce. VKAs were the first anticoagulants used in AF. Although these drugs are highly effective in the prevention of thromboembolism, their use is limited by their narrow therapeutic interval that necessitates frequent monitoring and dose adjustment. Also, multiple drug interactions and foods affect the metabolism of VKAs resulting in substantial risk and inconvenience. Multiple interactions and efforts to monitor and dose-adjust VKA therapy could have a direct impact on patient QoL and satisfaction. On the other hand, the use of DOACs in clinical practice is increasing rapidly, having a predictable effect without the need for narrow monitoring and a lower drug interaction profile. We have evaluated differences in QoL between a contemporary cohort of AF patients on VKAs and DOACs and patient satisfaction when taking them.

## 2. Methods

Patients in the REGUEIFA study (Registro Gallego Intercéntrico de Fibrilación Auricular) on anticoagulant treatment were included. The REGUEIFA study is a prospective, observational, multicentre registry of patients with a primary or secondary diagnosis of AF, as well as being more than 18 years old and diagnosed and treated by a cardiologist, from 8 hospitals of a Community Health Area (Galicia) in Northwest Spain. Patients were included from January 2018 to February 2020, and all completed the informed consent. A detailed description of the study design has previously been published [11]. Demographic characteristics and QoL were evaluated at inclusion time. The Registry was approved by the Ethical Clinical Investigation Committee from Galicia with the Register Code number 2016/376.

The EuroQol-5D (EQ-5D) questionnaire was used to evaluate QoL; it is an instrument that evaluates generic QoL developed in Europe and widely used [12]. The EQ-5D consists of five questions, one for each of five dimensions that include mobility, self-care, usual activities, pain/discomfort, and anxiety/depression with three severity levels for each: level 1 represents no problems, level 2 represents some problems, and level 3 represents extreme problems. The health state 11111 indicates no health problems in any dimension. EQ-5D answers can be converted into an EQ-5D index, ranging from 0 (death) to 1 (perfect health) [13]. The associated questionnaire includes the visual analogue scale (VAS), in which patients can report their perceived health with a grade from 0 (worst possible health) to 100 (best possible health).

The Anti-Clot Treatment Scale (ACTS) questionnaire measured the burden and benefits of patients on VKAs and DOACs (see Appendix A): it consists of 17 items and is a Likert-type scale with five possible answers (1 = none to 5 = a lot) [14]. The first 12 items assess the patient burden on anticoagulant therapy, ranging from 12 to 60, while items 14–16 assess treatment benefits, ranging from 3 to 15. Item 13 assesses the negative impact of anticoagulant treatment on daily life, while item 17 assesses the positive impact of treatment on patients’ lives. Scores for the burden scale (items 1–12) are considered inversely (from 1 = a lot to 5 = none). Higher scores on the burden and benefits scale indicate greater satisfaction with anticoagulation treatment (burden: the higher the score, the lower the burden; benefit: the higher the score, the greater the benefit).

### Statistical Analysis

The quantitative variables are described as mean ± standard deviation (SD), with qualitative variables given as percentages. The Mann–Whitney test was used to evaluate the differences between groups for continuous variables, and the Chi-squared test or the Fisher’s exact test were used for categorical variables.

A logistic regression model evaluated clinical factors related to EQ-5D dimensions and whether the OAC type was an independent predictor of EQ-5D dimensions. The model included all characteristics that differed between VKAs and DOACs, and the relevant baseline characteristics associated with QoL.

A linear regression model was used to evaluate clinical factors related to EQ-5D index and VAS score, and to evaluate if the OAC type was an independent predictor of these values. The model included all characteristics that differed between VKAs and DOACs, and relevant baseline characteristics associated with QoL.

## 3. Results

A total of 904 patients on anticoagulant treatment were included, as well as 532 patients on VKAs and 372 on DOACs. Patients on VKAs were older than those on DOACs (71.69 ± 10.35 vs. 63.61 ± 11.81, *p* < 0.001), with more women on VKAs than DOACs (37.03% vs. 29.57%, *p* < 0.001). Patients on VKAs had higher CHA2DS2-VASc and HASBLED scores (2.87 ± 1.4 vs. 1.93 ± 1.43, *p* < 0.001 and 0.91 ± 0.78 vs. 0.47 ± 0.66, *p* < 0.001, respectively) and comorbidity. Table 1 summarises baseline characteristics.

In addition, 846 patients completed the EQ-5D questionnaire, with 490 on VKAs and 356 on DOACs. Patients on VKAs had more mobility limitations than patients on DOACs (37.55% vs. 24.16%, *p* < 0.001), more restriction with usual activities (24.69% vs. 18.26%, *p* = 0.026), and more pain/discomfort (31.84% vs. 24.16%, *p* = 0.015). There were fewer patients with health status 11111 in the VKA group (39.39% vs. 55.34%, *p* < 0.001). Table 2 shows the results of the EQ-5D questionnaire. The VAS score was also lower in patients on VKAs (66.35 vs. 70.75, *p* < 0.001) (Table 2 and Figure 1).

Adjusting for baseline characteristics, the type of anticoagulant was not an independent factor for more mobility limitations, more restriction in usual activities, or more pain/discomfort (Table 3). Yet, female gender, coronary artery disease, HASBLED score, and symptomatic AF (EHRA scale ≥ 2) were associated with greater limitations. The type of anticoagulant was not an independent predictor of the EQ-5D index, as was the VAS score in a multivariate analysis (Table 4). Clinical factors (female gender, chronic obstructive pulmonary disease (COPD), higher CHA2DS2-VASc scores, EHRA scale ≥ 2, and diuretic use) were linked to a lower EQ-5D index and VAS score.

In addition, 713 patients completed the ACTS questionnaire, along with 416 on VKAs, and 297 on DOACs. Patients on VKAs had lower scores for burden and benefit than those on DOACs (52.1 vs. 55.5, *p* < 0.001 and 11.13 vs. 11.75, *p* < 0.001, respectively), which indicates lower satisfaction with anticoagulation treatment (Table 5). The negative impact of anticoagulant treatment on daily life was higher for VKAs, and the positive impact of anticoagulant treatment was also lower for VKAs (Table 5).

## 4. Discussion

This study includes a contemporary cohort of AF patients on OAC treatment and evaluates differences in QoL and patient satisfaction while taking VKAs or DOACs.

Main findings: (1) Patients on VKAs had more mobility limitations, more restriction in usual activities, and more pain/discomfort, as well as a lower EQ-5D index and VAS scores than patients on DOACs; (2) baseline characteristics differed between taking VKAs and DOACs; (3) patient satisfaction with anticoagulant treatment improved on DOACs.

It is well-known that QoL is impaired in AF patients, independent of other cardiovascular conditions [7,8,9]. There are limited data comparing the benefit of different anticoagulant treatments regarding QoL. Moreover, QoL assessment is constrained by the presence of different tools to assess it, and the lack of cross-validation of several AF-specific QoL tools [7,15,16,17]. We used the EQ-5D questionnaire to evaluate QoL in AF patients receiving OAC, VKAs, and DOACs, although we found more mobility limitations, more restriction in usual activities, and more pain/discomfort in patients taking VKAs. We also found a lower EQ-5D index and a lower VAS score in those on VKAs. However, when analysing baseline characteristics, we found more comorbidity in patients on VKAs, as it was not an independent factor for worse QoL; female gender, CHADS-VASc and HASBLED scores, coronary artery disease, COPD, and EHRA classification were associated with more QoL limitations. These findings are assessed according to the literature, which consistently describes poorer QoL in female AF patients, but it is not clear if this reflects gender differences in the population or is related to AF per se [18]. The EHRA scale describes symptom severity in AF patients, and is related to QoL [19,20]. Several patient factors, such as symptomatic heart failure, diabetes, COPD, and coronary artery disease, are associated with worse QoL [9,21].

Baseline clinical characteristics are significantly different between anticoagulants, VKAs, and DOACs, and these variations are associated with patient QoL, as we mentioned before. The marked differences between patients in this study, in which patients were included and treated by a cardiologist, can be explained by the restriction to prescribe DOACs, established by our health system [22]. Briefly, our National Health System considers the use of DOACs in patients with allergy or specific contraindication for VKA, in those with previous intracranial haemorrhage, in patients with a stroke and high risk of haemorrhagic transformation, in patients with arterial embolism, despite use of VKA with good range control, and those on VKA with poor controls (time of therapeutic range < 60% for 6 months). In all other cases, our National Health System does not finance prescriptions for DOAC. The only way to prescribe them, excluding previous criteria, involves intervention (electrical cardioversion or ablation), in which the need for OAC lasts 2–3 months without chronic anticoagulation; in these cases, the hospital pharmacy provides patient treatment. The prescription policies in our country could contribute to the fact that the population of patients treated with DOACs in our series being selected for a less unfavourable risk profile, thus conditioning our results. In these regard, VKA patients are older, are more often women, and have more comorbidity and risk factors. Also, in our study, the number of patients with DOACs was lower than in other series and the catheter ablation patients are underrepresented (14%). However, in another state perspective, an Italian multicentre, non-interventional, prospective study that enrolled 250 consecutive atrial fibrillation patients eligible for catheter ablation on rivaroxaban demonstrated the safety and efficacy of rivaroxaban uninterrupted or shortly interrupted prior to catheter ablation [23].

Finally, we used the ACTS questionnaire, a validated metric of patient-reported burden and benefit of oral anticoagulation, to evaluate patient satisfaction with anticoagulant treatment, which was higher for DOACs than for VKA. Higher ACTS scores for patients on DOACs vs. VKA were previously reported in other cohorts [24,25]. These findings are likely due to differences between each treatment. VKA was the mainstay of OAC but is a difficult drug to dose and monitor. Due to the variability in dose-response with VKA treatment, monitoring the degree of anticoagulation is imperative, usually needing a changed dose [26]. Patients on VKA must follow a strict diet, due to multiple food interactions and drug–drug interactions. However, DOACs have major pharmacologic advantages over VKA, including rapid onset and offset of action, few drug interactions, and predictable pharmacokinetics, eliminating the need for regular coagulation monitoring [27]. All these advantages may justify the greater satisfaction found in patients on DOACs. This information, provided by ACTS questionnaire, suggests that for patients taking VKA, patient-reported care satisfaction should be routinely assessed, and, in those patients with low satisfaction, switching from VKA to DOACs should be considered.

## 5. Limitations

This is an observational study; however, it included a high number of patients from different regions, so findings on baseline characteristics, QoL, and patient satisfaction with treatment reflect the data of a wide AF population.

The EQ-5D questionnaire is generic and not designed specifically for AF patients; such that its sensitivity and discriminative capacity is lower than that of other questionnaires. It has been described as a ceiling effect (patients with maximum scores and health status 11111), which may present some difficult comparisons.

The ACTS questionnaire does not provide information about patient satisfaction, so some issues may not be represented as outcomes, self-efficacy, or patient acceptance of treatment. Other factors, such as the duration of OAC therapy before administering the ACTS questionnaire could influence results. In this first study, we did not evaluate whether the burden and benefit scores changed during follow-up.

## 6. Conclusions

In a contemporary cohort of AF patients taking OAC, those on VKAs had more comorbidity, more mobility limitations, more restriction in usual activities and more pain/discomfort, and a lower EQ-5D index and VAS scores than patients on DOACs. VKA has a greater burden and higher negative impact on the patient’s life than DOACs.

## Figures and Tables

**Figure 1 jcm-13-05283-f001:**
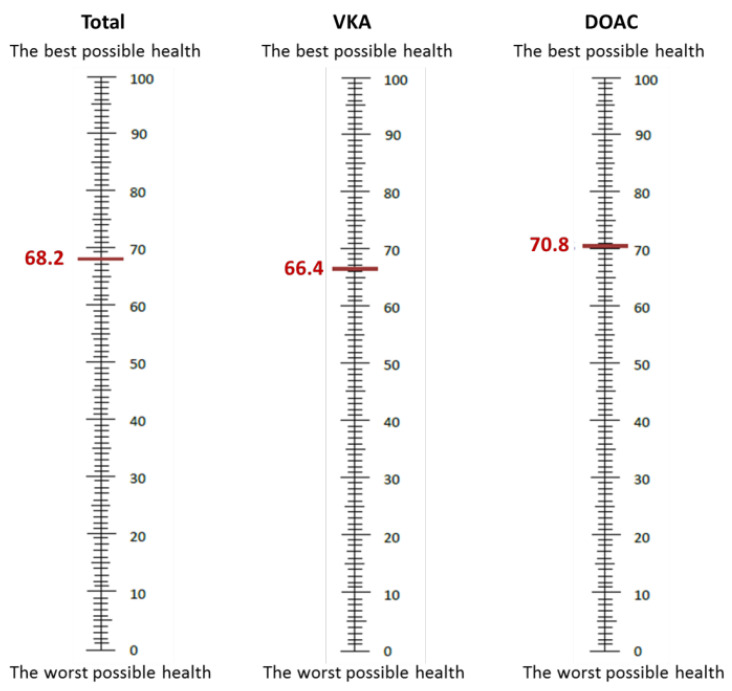
Visual analogue scale scores.

**Table 1 jcm-13-05283-t001:** Baseline characteristics.

Variables	Total (*n* = 904)	VKA (*n* = 532)	DOAC (*n* = 372)	*p*-Value
Age, median (IQR)	69 (61–77)	72 (65–79)	64 (56–71.5)	<0.001
Women, %	33.96	37.03	29.57	0.020
Renal failure, %	6.86	10.34	1.88	<0.001
PAD, %	2.99	4.14	1.34	0.016
CAD, %	12.17	14.29	9.14	0.020
Dementia, %	0.66	1.13	0.00	0.046
OSAHS, %	4.87	4.89	4.84	0.973
COPD, %	11.28	13.35	8.33	0.019
Anaemia, %	3.98	5.45	1.88	0.009
Diabetes, %	19.91	22.74	15.86	0.011
Hyperlipidaemia, %	50.00	53.57	44.89	0.010
Hypertension, %	66.37	72.93	56.99	<0.001
Obesity (BMI ≥ 30), %	42.29	41.24	43.78	0.448
CHA2DS2-VASc score, median (IQR)	2 (1–4)	3 (2–4)	2 (1–3)	<0.001
HAS-BLED score, median (IQR)	1 (0–1)	1 (0–1)	0 (0–1)	<0.001
EHRA ≥ II, %	66.48	65.41	68.01	0.416
AF classification				<0.001
First diagnosed episode, %	23.12	25.56	19.62	
Paroxysmal, %	20.24	17.86	23.66	
Persistent, %	27.43	21.80	35.48	
Long standing persistent, %	2.54	2.26	2.96	
Permanent, %	26.66	32.52	18.28	
EF < 50%, %	18.30	19.55	16.57	0.268
Valvular heart disease, %	17.26	21.05	11.83	<0.001
Beta blockers, %	68.47	65.98	72.04	0.053
Calcium channel blockers, %	5.64	6.02	5.11	0.561
Digoxin, %	6.64	7.89	4.84	0.069
ACE inhibitors/ARBs, %	54.20	54.89	53.23	0.622
Diuretics, %	36.06	42.67	26.61	<0.001
Antiarrhythmic drugs, %	38.83	28.01	54.30	<0.001

Abbreviations: IQR, interquartile range; PAD, peripheral artery disease; CAD, coronary artery disease; OSAHS, obstructive sleep apnoea–hypopnea syndrome; COPD, chronic obstructive pulmonary disease; BMI, body mass index; EHRA, European Heart Rhythm Association score; EF, ejection fraction; ACE, angiotensin converting enzyme; ARBs, angiotensin II receptor blockers.

**Table 2 jcm-13-05283-t002:** EQ-5D questionnaire scores.

	Total (*n* = 846)	VKA (*n* = 490)	DOAC (*n* = 356)	*p*-Value
Mobility				
Patients with some problems (%)	31.91	37.55	24.16	<0.001
Patients with extreme problems (%)	0.47	0.41	0.56	1.000
Self-care				
Patients with some problems (%)	7.92	8.78	6.74	0.279
Patients with extreme problems (%)	0.71	0.61	0.84	0.700
Usual activities				
Patients with some problems (%)	21.99	24.69	18.26	0.026
Patients with extreme problems (%)	1.89	2.04	1.69	0.802
Pain/Discomfort				
Patients with some problems (%)	28.61	31.84	24.16	0.015
Patients with extreme problems (%)	2.72	3.47	1.69	0.136
Anxiety/depression				
Patients with some problems (%)	26.12	26.53	25.56	0.751
Patients with extreme problems (%)	3.19	3.88	2.25	0.235
Visual Analog Scale (VAS), mean ± SD	68.2 ± 16.1	66.35 ± 16.21	70.75 ± 15.62	<0.001
EQ-5D index, mean ± SD	0.82 ± 0.21	0.79 ± 0.21	0.85 ± 0.2	<0.001
Patients with health status 11111 (%)	46.10	39.39	55.34	<0.001

**Table 3 jcm-13-05283-t003:** Multivariate logistic regression analysis.

**Mobility**				
	**Final Model**			
	**OR**	**CI 95%**		***p*-Value**
OAC (DOAC)	0.99	0.69	1.42	0.959
Age	1.04	1.02	1.07	<0.001
Female	2.46	1.71	3.55	<0.001
CAD	1.84	1.13	2.98	0.014
COPD	1.91	1.15	3.20	0.013
CHA2DSVASc score	1.2	1	1.43	0.044
HASBLED score	1.2	1	1.43	0.006
EHRA scale ≥ 2	1.75	1.20	2.56	0.004
AF type				
Paroxysmal	1.41	0.82	2.42	0.212
Persistent	2.42	1.42	4.14	0.001
Long-standing persistent	2.43	0.84	7.06	0.102
Permanent	1.59	0.96	2.63	0.069
Diuretics	1.57	1.09	2.27	0.015
**Usual Activity**				
	**Final Model**			
	**OR**	**CI 95%**		** *p* ** **-Value**
OAC (DOAC)	0.99	0.68	1.44	0.966
Female	2.01	1.41	2.88	<0.001
CAD	2.05	1.27	2.31	0.003
HASBLED score	1.70	1.32	2.19	<0.001
EHRA scale 3–4	4.27	2.40	7.62	<0.001
AF type				
Paroxysmal	2.70	1.48	4.93	0.001
Persistent	2.81	1.54	5.11	0.001
Long-standing persistent	2.04	0.63	6.55	0.233
Permanent	2.80	1.59	4.94	<0.001
Diuretics	1.77	1.22	2.58	0.003
**Pain/Discomfort**				
	**Final Model**			
	**OR**	**CI 95%**		** *p* ** **-Value**
OAC (DOAC)	0.97	0.69	1.36	0.854
Female	1.95	1.39	2.74	<0.001
CAD	1.57	0.99	2.48	0.055
CHA2DSVASc score	1.25	1.08	1.44	0.003
HASBLED score	1.28	1	1.68	0.047
EHRA scale 3–4	2.42	1.43	4.11	0.001
Valvular heart disease	1.51	1.01	2.27	0.046

Abbreviations as Table 1.

**Table 4 jcm-13-05283-t004:** Multivariate linear regression analysis.

**EQ-5D Index**				
	**Final Model**			
	**Coefficient**	**CI 95%**		***p*-Value**
OAC (DOAC)	0.009	−0.019	0.036	0.534
Female	−0.092	−0.121	−0.063	<0.001
CAD	−0.055	−0.095	−0.016	0.006
COPD	−0.059	−0.101	−0.017	0.006
CHA2DSVASc score	−0.014	−0.026	−0.001	0.029
HASBLED score	−0.040	−0.061	−0.019	<0.001
EHRA scale ≥ 2	−0.040	−0.068	−0.013	0.004
AF type				
Paroxysmal	−0.026	−0.065	0.013	0.185
Persistent	−0.039	−0.076	−0.001	0.042
Long-standing persistent	−0.041	−0.123	−0.042	0.336
Permanent	−0.044	−0.082	−0.006	0.024
Diuretics	−0.039	−0.068	−0.009	0.011
**VAS**				
	**Final model**			
	**Coefficient**	**CI 95%**		** *p* ** **-Value**
OAC (DOAC)	1.47	−0.74	3.68	0.192
Female	−3.39	−5.73	−1.05	0.005
COPD	−4.31	−7.68	−0.93	0.012
Anaemia	−5.68	−11.03	−0.33	0.038
Hyperlipidaemia	−2.91	−5.03	−0.79	0.007
CHA2DSVASc score	−1.09	−2	−0.19	0.018
EHRA scale ≥2	−5.10	−7.33	−2.86	<0.001
Diuretics	−2.80	−5.18	−0.41	0.021
Antiarrhythmic drug treatment	2.44	0.16	4.72	0.036

Abbreviations as Table 1.

**Table 5 jcm-13-05283-t005:** ACTS questionnaire scores.

	Total (*n* = 713)	VKA (*n* = 416)	DOAC (*n* = 297)	*p*-Value
Burden score, mean ± SD	53.51 ± 7.96	52.09 ± 8.44	55.5 ± 6.75	<0.001
Benefit score, mean ± SD	11.39 ± 2.51	11.13 ± 2.42	11.75 ± 2.6	<0.001
General negative impact, mean ± SD	1.68 ± 1.01	1.77 ± 1.02	1.56 ± 0.99	<0.001
Distribution of scores, %				<0.001
1	58.20	51.20	68.01	
2	26.51	31.73	19.19	
3	6.59	8.65	3.70	
4	6.17	5.29	7.41	
5	2.52	3.13	1.68	
General positive impact, mean ± SD	3.65 ± 0.99	3.56 ± 0.96	3.78 ± 1.02	<0.001
Distribution of scores, %				<0.001
1	3.65	3.61	3.70	
2	9.96	10.34	9.43	
3	20.62	25.00	14.48	
4	49.51	49.04	50.17	
5	16.27	12.02	22.22	

Burden: the higher the score, the lower burden. Benefit: the higher the score, the greater the benefit.

## Data Availability

Data supporting the reported results can be found in a dataset of REGUEIFA Study (Odds S.L., A Coruña, Spain) and can be consulted upon request.

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
