# Peer review of "Health-Related Quality of Life and Satisfaction in Atrial Fibrillation Patients on Anticoagulant Therapy: Differences between Vitamin K Antagonists and Direct Oral Anticoagulants; Results from the Multicentre REGUEIFA Registry"

_jcm, 2024, doi:10.3390/jcm13175283_

Round 1
Reviewer 1 Report
Comments and Suggestions for Authors
Congratulations to authors for considering this aspect of oral anticoagulation from the perspective of a state. The paper is interesting and well conducted. However I have some comments that I think could improve your manuscript:
- The abstract should be clearer in the results section and should be more fluent.
- There are some typo in the text that must be corrected
- In the study it is mentioned that the use of DOACs is restricted by the National Health System, and this could potentially biase the sample toward patients with specific indications for DOACs over VKAs. Therefore the authors should discuss how these prescription policies may impact the study findings.
- In order to empower your background, I strongly suggest to include and discuss another state perspective: “Italian Registry in the Setting of Atrial Fibrillation Ablation with Rivaroxaban - IRIS. Minerva Cardiol Angiol. 2024 May 30. doi: 10.23736/S2724-5683.24.06546-3. Epub ahead of print. PMID: 38814252.”
- The statistic section is properly done; anyway authors should consider hierarchical data structures as well as using mixed-effects models to account for intra-patient variability.
Comments on the Quality of English Language
the english form must be improved especially regarding typo errors
Author Response
Reviewer 1:
Congratulations to authors for considering this aspect of oral anticoagulation from the perspective of a state. The paper is interesting and well conducted. However I have some comments that I think could improve your manuscript:
The abstract should be clearer in the results section and should be more fluent”.
Response: We thank Reviewer 1 for taking the time to review our work. We have reviewed and improved the abstract in the results section and globally to be more fluent as suggested.
There are some typos in the text that must be corrected”.
Response: We appreciate the Reviewer comment. We have included a manuscript English and grammar revision as suggest.
In the study it is mentioned that the use of DOACs is restricted by the National Health System, and this could potentially bias the sample toward patients with specific indications for DOACs over VKAs. Therefore, the authors should discuss how these prescription policies may impact the study findings".
Response: We appreciate the Reviewer comment. We have added the last sentence in the discussion section as suggested.
"These prescription polices in our country could contribute to the fact that the population of patients treated with DOACs in our series being selected for a less unfavourable risk profile, thus conditioning our results".
In order to empower your background, I strongly suggest to include and discuss another state perspective: “Italian Registry in the Setting of Atrial Fibrillation Ablation with Rivaroxaban - IRIS. Minerva Cardiol Angiol. 2024 May 30. doi: 10.23736/S2724-5683.24.06546-3. Epub ahead of print. PMID: 38814252."
Response: We have added the reference as suggested and discuss another state perspective in Europe in relation with the data of Italian Registry in the Setting of Atrial Fibrillation Ablation with Rivaroxaban. We have added the last sentences and in the discussion section.
“The prescription polices in our country could contribute to the fact that the population of patients treated with DOACs in our series being selected for a less unfavourable risk profile, thus conditioning our results. In these regard, VKA patients are older, have more women, and more comorbidity and risk factors. Also, in our study, the number of patients with DOACs was lower than in other series and the catheter ablation patients are underrepresented (14%). However, in another state perspective, an Italian multicenter, non-interventional, prospective study which enrolled 250 consecutive atrial fibrillation patients eligible for catheter ablation on rivaroxaban demonstrated the safety and efficacy of rivaroxaban uninterrupted or shortly interrupted prior to catheter ablation”.23
(Reference 23) Lavalle C, Pierucci N, Mariani MV, Piro A, Borrelli A, Grimaldi M, Rossillo A, Notarstefano P, Compagnucci P, Dello Russo A, Perna F, Pelargonio G, LA Fazia VM, Della Rocca DG, Miraldi F, Forleo GB. Italian Registry in the Setting of Atrial Fibrillation Ablation with Rivaroxaban - IRIS. Minerva Cardiol Angiol. 2024 May 30. doi: 10.23736/S2724-5683.24.06546-3. Epub ahead of print.
The statistic section is properly done; anyway authors should consider hierarchical data structures as well as using mixed-effects models to account for intra-patient variability”.
Response: We thank Reviewer for his suggestion regarding hierarchical data structures as well as using mixed-effects models to account for intra-patient variability in the analysis. This specific analysis was discarded for the following reasons.
We analyzed in anticoagulated patients whether the type of anticoagulant is related to quality of life (with different dimensions of it). Since giving one or another anticoagulant is predictably related to patient characteristics that, in addition, can also be related to quality of life, we try to make a broad adjustment (in terms of patient characteristics). The reviewer's suggestion of hierarchical analysis implies assuming that there are factors at different levels that influence quality of life. The truth is that I can't think of any other level that could be suggesting other than the hospital, since it will be difficult to assign a patient to a single doctor, which could be another possible level. It was not considered as a hypothesis that the quality of life in these patients depends on where they are treated and that is the reason for not considering this analysis. I believe that the greatest justification for discarding this analysis is the organizational similarity of all the centers (public system, similar healthcare access, identical financing of medication, etc.). In addition, for there to be measurable intra-patient variability, there must be more than one measurement in each patient, something that does not happen in our study. These models are very appropriate for situations of multiple measurements over time in a patient cohort, which is not the case of our study.
Comments on the Quality of English Language: the English form must be improved especially regarding typo errors.
Response: We have included a manuscript English revision and thank the Manager Editor for this suggestion.

Reviewer 2 Report
Comments and Suggestions for Authors
The present study is an extremely interesting one, however the following must be taken into account:
1. Lines 46-47 reference missing.
2. For introduction, please provide a more detailed comparison between the two classes of anticoagulants. There are more important references (like ARISTOTLE) that suppose the fact the also DOACs are good for stroke prevention, than those provided in the manuscript. Please rewrite the introduction up to one page.
3. Please provide the NCT number of the clinical trial or ethics committee approval.
4. It is unclear whether the observational study was prospective or retrospective, please specify in the materials and methods section.
5. Were all the patients included in the study from Spain, or other countries participated in the patient’s inclusion also? Please specify accordingly.
6. Since you use Mann-Whitney test to evaluate the differences between groups, why did you reported the results as mean ± SD and not median [IQ25-IQ75]. After all Mann Whitney is a test comparing medians, not means.
Comments on the Quality of English Language
Author Response
Reviewer 2:
The present study is an extremely interesting one, however the following must be taken into account:”
Lines 46-47 reference missing”.
Response: We thank Reviewer 1 for taking the time to review our work. We apologize for the error. We have included the references missing as suggested.
(Reference 1) Chugh SS, Havmoeller R, Narayanan K, Singh D, Rienstra M, Benjamin EJ, Gillum RF, Kim YH, McAnulty JH Jr, Zheng ZJ, Forouzanfar MH, Naghavi M, Mensah GA, Ezzati M, Murray CJ. Worldwide epidemiology of atrial fibrillation: a Global Burden of Disease 2010 Study. Circulation 2014;129:837–847.
(Reference 2) Andersson T, Magnuson A, Bryngelsson IL, Frobert O, Henriksson KM, Edvardsson N, Poci D. All-cause mortality in 272,186 patients hospitalized with incident atrial fibrillation 1995-2008: a Swedish nationwide long-term casecontrol study. Eur Heart J 2013;34:1061–1067.
(Reference 3) Wolf PA, Abbott RD, Kannel WB. Atrial fibrillation as an independent risk factor for stroke: the Framingham Study. Stroke 1991;22:983–988.
For introduction, please provide a more detailed comparison between the two classes of anticoagulants. There are more important references (like ARISTOTLE) that suppose the fact the also DOACs are good for stroke prevention, than those provided in the manuscript. Please rewrite the introduction up to one page”.
Response: We thank Reviewer for his suggestion. We rewrite the introduction up to one page including a more detailed comparison between VKA and DOACs. In addition, we included the reference of ARISTOTLE trial as suggested.
(Reference 6) Granger CB, Alexander JH, McMurray JJ, Lopes RD, Hylek EM, Hanna M, Al-Khalidi HR, Ansell J, Atar D, Avezum A, Bahit MC, Diaz R, Easton JD, Ezekowitz JA, Flaker G, Garcia D, Geraldes M, Gersh BJ, Golitsyn S, Goto S, Hermosillo AG, Hohnloser SH, Horowitz J, Mohan P, Jansky P, Lewis BS, Lopez-Sendon JL, Pais P, Parkhomenko A, Verheugt FW, Zhu J, Wallentin L; ARISTOTLE Committees and Investigators. Apixaban versus warfarin in patients with atrial fibrillation. N Engl J Med. 2011 Sep 15;365(11):981-92.
Please provide the NCT number of the clinical trial or ethics committee approval”.
Response: We thank Reviewer for his suggestion. We included the ethics committee approval as suggested.
“The Registry was approved by the Ethical Clinical Investigation Committee from Galicia (CEI) with the Register Code number 2016/376”.
It is unclear whether the observational study was prospective or retrospective, please specify in the materials and methods section”.
Response: We have added the information requested in the materials and methods section. The REGUEIFA study is a prospective, observational and multicenter registry.
Were all the patients included in the study from Spain, or other countries participated in the patient’s inclusion also? Please specify accordingly”.
Response: We thank Reviewer for his suggestion. We included the information of the registry. All patients were included in Spanish hospitals.
We have added the last sentence and in the discussion section as suggested.
“The REGUEIFA study is a prospective, observational, multicenter registry of patients with a primary or secondary diagnosis of AF, as well as being more than 18-years-old, and diagnosed and treated by a cardiologist, from 8 hospitals of a Community Health Area (Galicia) in Northwest of Spain”.
Since you use Mann-Whitney test to evaluate the differences between groups, why did you reported the results as mean ± SD and not median [IQ25-IQ75]. After all Mann Whitney is a test comparing medians, not means”.
Response: We thank Reviewer for his suggestion. We change mean ± SD by median [IQ25-IQ75] as suggested.
Comments on the Quality of English Language: the English form must be improved especially regarding typo errors.
Response: We have included a manuscript English revision and thank the Manager Editor for this suggestion.

Round 2
Reviewer 1 Report
Comments and Suggestions for Authors
Congratulations to author for the revised version of the manuscript. You managed to improve it so remarkably.